# Associations of Spiritual Well-Being and Hope with Health Anxiety Severity in Patients with Advanced Coronary Artery Disease

**DOI:** 10.3390/medicina57101066

**Published:** 2021-10-06

**Authors:** Rasoul Heshmati, Eisa Jafari, Tahereh Salimi Kandeh, Marie L. Caltabiano

**Affiliations:** 1Department of Psychology, Faculty of Education and Psychology, University of Tabriz, Tabriz 5166, Iran; 2Department of Psychology, Faculty of Education and Psychology, Payame Noor University, Tehran 4697, Iran; Salimi@pnu.ac.ir; 3Department of Psychology, College of Healthcare Sciences, Division of Tropical Health & Medicine, James Cook University, Cairns 4870, Australia; marie.caltabiano@jcu.edu.au

**Keywords:** health anxiety, spiritual well-being, advanced coronary artery disease

## Abstract

*Background and Objectives:* Health anxiety is one of the most common problems in patients with coronary artery disease. The present study tested whether health anxiety severity could be predicted by spiritual well-being and hope in patients with advanced coronary artery disease. *Materials and Methods*: In a cross-sectional study, 100 patients with advanced coronary artery disease were recruited from hospitals and healthcare centers in Iran. Patients completed self-report scales, including the Spiritual Well-Being Scale, Adult Hope Scale, and Short Health Anxiety Inventory. Hierarchical multiple regression analyses were used to empirically explore the relations among variables. *Results:* Results indicated that patients who reported higher levels of hope (β = 0.42, *p* < 0.01) and spiritual well-being (β = 0.20, *p* < 0.05) reported lower levels of health anxiety. Agency (β = 0.58, *p* < 0.01) scores were a significant negative predictor of health anxiety severity. Additionally, religious spirituality scores (β = 0.28, *p* < 0.01) were shown to significantly negatively predict health anxiety level. However, the pathways components of hope and existential spirituality were not significant predictors. *Conclusion:* The findings of the present study indicate that spiritual well-being and hope could be important factors in determining health anxiety for adults with coronary artery disease, and their role is worthy of further exploration to help improve health anxiety for patients with coronary artery disease.

## 1. Introduction

### 1.1. Background

Coronary artery disease (CAD), also known as ischemic heart disease, is the most common of the cardiovascular diseases. CAD is caused by a buildup of plaque on the walls of arteries that supply blood to the heart and other parts of the body. Despite significant advances in the diagnosis and treatment of this disease, it is still one of the leading causes of death worldwide [1]. In the United States and Canada, more than 6% of adults older than 20 years of age have CAD as the most common form of cardiovascular disease [2]. Epidemiological studies in Iran also show that CAD is one of the leading causes of death and disability of individuals. The Iranian Ministry of Health and Medical Education (MOHME) reported that 39.03% of all deaths in Iran are due to CAD [3]. Most studies on the role of psychological factors in heart diseases have confirmed the relationship between psychological factors such as emotional distress and cardiovascular disease [4]. Psychological distress is more prevalent and severe in patients with CAD than in healthy individuals [5]. There is evidence indicating that the course of the disease and its complications, along with the need for invasive procedures in some cases, leads to psychological problems such as anxiety in patients. Anxiety is one of the most common psychological reactions to cardiovascular disease [6,7]. Moreover, it is one of the significant psychological symptoms with an estimated prevalence of 20–50% in adults with CAD [8]. In other words, anxiety is a comorbidity in CAD patients [9], and its symptoms are present in most of the patients. One type of anxiety that may manifest itself in chronic medical conditions, such as coronary heart disease, is health anxiety.

Health anxiety is worrying about physical symptoms believed to indicate a serious illness [10,11]. Health anxiety is a continuum with mild worries about physical sensations at one end and severe worries of health and mental occupation with physical sensations at the other end [12]. It can occur at any time during the life of a person with CAD. Research suggests that health anxiety is significantly associated with severe clinical conditions [13]. For example, research has shown that children and adolescents with Congenital Heart Defects (CHDs) experience high levels of health anxiety compared to normal children and adolescents [14]. Another study found that individuals with CHD experience health anxiety [15]. Numerous psychological factors, including spiritual well-being, can affect health anxiety in patients with coronary heart disease.

Spiritual well-being is defined as a sense of connection with others, having meaning and purpose in life, and believing in and relating to a superior or higher power [16]. Spiritual well-being has two dimensions: religious and existential well-being. Religious well-being is one’s relationship with God and religion, while existential well-being is related to one’s mental well-being regarding the meaning and purpose of life, as well as life satisfaction [17]. Although there is no research examining the predictive role of spiritual well-being in the health anxiety of patients with CAD, several studies have examined the role of spiritual well-being in the context of life-threatening illnesses. For example, one study investigated the relationship between spiritual well-being and its components with death anxiety in a group of elderly people during the COVID-19 pandemic, and the results showed that spiritual and religious well-being predicted death anxiety in these individuals [18]. Another study showed that with decreasing spiritual well-being scores, depression, anxiety, and stress increase in patients with heart failure and in family caregivers [19]. On the other hand, research shows that hope plays an important role in reducing anxiety in people with coronary heart disease.

Hope is a kind of thought process and consists of two components of agency (purposeful wills) and pathways (purposeful plans), both of which are necessary in the formation and determination of hope through purposeful behaviors. Agency is the motivating component of hope and reflects a personal perception of the ability to achieve past, present, and future goals. The pathway is a cognitive component of hope and reflects an individual’s ability to create sensible ways to achieve goals [20]. Hope can be considered as an active and dynamic state of existential coping among patients with life-threatening illness. The role of hope in the level of health anxiety of patients with CAD has not been studied, and most studies have addressed the role of hope in anxiety, depression, and adjustment of patients. One study investigated the relation between hope, anxiety and health-related quality of life in a number of children and adolescents with cancer, and the results showed that there was a positive relation between hope and health-related quality of life [21]. Hope was directly and indirectly associated with reduced anxiety in children and adolescents with cancer [21]. Research evidence also shows that there is a relation between hope and life satisfaction, psychological adjustment, and coping strategies in patients with spinal cord injury [22]. In a cross-sectional study, the relation between hope and anxiety was examined in a group of patients who had heart surgery; there was a negative and significant relation between life expectancy and anxiety in these patients [23]. In another study, the association between hope and anxiety in individuals with advanced heart disease was investigated, and the results indicated that hope had a negative correlation with anxiety in these patients [9]. 

### 1.2. Objective

A review of the literature indicates that previous studies have paid less attention to the predictive role of spiritual well-being and hope in the health anxiety of coronary heart disease patients, especially in Iran, as a religious country where spirituality and religious concepts can play an important role in helping individuals deal with the disease and health anxiety; however, the role of these variables in the health anxiety of CAD patients is not clear. In order to overcome the gap in the research area and the necessity and importance of identifying the variables of spiritual well-being and hope in the health anxiety of patients with advanced CAD, the present study aims to address the following question: Is there a relationship between spiritual well-being and hope with health anxiety in patients with advanced CAD?

## 2. Materials and Methods

### 2.1. Participants and Procedure

To investigate the predictive role of spiritual well-being and hope on health anxiety in CAD patients, this exploratory study used a quantitative, descriptive–correlational, cross-sectional design. One hundred and eight patients with coronary artery disease who were referred to hospitals and healthcare centers in Ardabil city (Iran) were recruited from October to December 2020. The eligibility criteria included the diagnosis of advanced CAD and an age range of 20 to 65 years. Exclusion criteria included history of attending meditation and yoga courses, experience of psychotherapy, a history of psychotic disorders, a history of surgery in the last 6 months, death of a family member in the last year, or experience of divorce in the last year. A semi-structured interview was undertaken to assess these criteria. Based on this assessment, eight participants were not eligible to participate in the study and were excluded. Of the sample, there were 65 males (64%) and 35 females (36%). The patients ranged in age from 23 to 65 years with a mean (SD) age of 44.53 years (7.62). The sample composition of participants’ highest level of education completed was 10.0% graduate school, 2.0% four years of college, 3.0% two years of college, 39% high school, and 46% less than high school. 

The research team informed participants about the aims of the research and received written consent by them. Then, they filled out the questionnaires. The patients were informed that the measures sought their actual feelings, thoughts, behaviors, and experiences in daily life, and they were encouraged to fill them out honestly and completely. The instruments took the patients about 40 minutes to complete. The time interval for administering the questionnaire among patients was 4 weeks.

### 2.2. Measures 

*Spiritual Well-Being Scale (SWBS):* Paloutzian and Ellison’s [24] SWBS is a general index of subjective well-being and perceived spiritual quality of life that includes 20 positive and negative items. Each item is scored on a six-point Likert scale, from “strongly disagree” (score 1) to “strongly agree” (score 6). The SWBS includes two subscales: “religious well-being (RWB)” and “existential well-being (EWB)” [25,26]. The RWB subscale consists of 10 odd-numbered items, which assess the individual’s relationship with God (i.e., “I believe that God loves me and cares about me”). The EWB subscale consists of 10 even-numbered items, which refer to the horizontal dimension of well-being about the world around us (i.e., “I don’t know who I am, where I came from, or where I’m going”). The total SWBS score is calculated by adding the RWB and EWB scores, which range from 20 to 120. Higher scores indicate a higher level of well-being. Construct validity for the SWBS in an Iranian sample has been reported, and Cronbach alpha coefficients of 0.81, 0.84, and 0.89 for the RWB, EWB, and SWBS, respectively [27]. In the present research, the Cronbach alpha coefficients for RWB, EWB, and SWBS were 0.81, 0.89, and 0.86, respectively.

*The Adult Hope Scale (AHS):* Snyder et al. developed the Adult Hope Scale, which includes a 12-item measure of trait hope [28]. The AHS includes two subscales: “Agency” and “Pathways”. Four items assess Agency (e.g., “I energetically pursue my goals”); 4 items evaluate Pathways (e.g., “I can think of many ways to get out of a jam”); and four distractor items are included which are not scored. Agreement with each item is rated on an eight-point Likert scale from 1 (definitely false) to 8 (definitely true). The total AHS score is calculated by adding the Agency and Pathway scores, which range from 8 to 64, with higher scores indicating greater levels of hope. According to research conducted on undergraduate samples, the AHS has acceptable levels of internal consistency (Cronbach’s alpha = 0.74–0.84) [28] and temporal reliability (10- week test–retest reliability r = 0.76–0.82) [28]. Kermani et al. reported appropriate Cronbach’s alpha coefficients of 0.77, 0.79, and 0.86 for the Agency, Pathway, and AHS, respectively, in an Iranian sample [29]. The Cronbach’s alpha coefficients for Agency, Pathway, and AHS in this study were 0.83, 0.88, and 0.85, respectively.

*The Short Health Anxiety Inventory (SHAI):* The Short Health Anxiety Inventory is an 18-item self-report tool based on the cognitive model of health anxiety [30], which measures health anxiety over the past 6 months (e.g., “Thoughts of illness are so strong that I no longer even try to resist them”). Participants are asked to rate each item with four response options to examine cognitive and behavioral aspects of health anxiety (e.g., 0 = “I do not worry about my health;” 1 = “I occasionally worry about my health;” 2 = “I spend much of my time worrying about my health;” 3 = “I spend most of my time worrying about my health”). The total score can range from 0 to 54, with higher scores indicating greater severity of health anxiety. The SHAI has acceptable psychometric properties including good internal consistency, test–retest reliability, and construct validity, and it is sensitive to treatment [31,32]. In an Iranian sample, Nargesi et al. reported an acceptable Cronbach’s alpha coefficient of 0.75 for the SHAI [33]. In the present study, the Cronbach’s alpha coefficient was excellent (alpha = 0.88).

### 2.3. Ethical Consideration 

This research was approved by the Bioethics committee of University of Tabriz (The Ethical code: IR.ARUMS.REC.1400.144, Approved date: 1 November 2020). The study’s objective and procedure were explained to the patients by the research team. Participants were informed that their participation was completely voluntary and that they had the right to withdraw from the study at any time without penalty. The participants were told that their answers and scores would be kept anonymous.

### 2.4. Data Analysis

Statistical analysis was conducted using the SPSS 24.0 version program. Since less than 5% of data for any variable was missing, it was decided that the missing data could be overlooked without having a significant impact on the analyses [34]. For participants with missing data, pairwise deletion was utilized. Descriptive statistics were conducted with mean, standard deviation (SD), number (N), and percentage (%) for study variables. Skewness and kurtosis were used to check that variables were normally distributed (acceptable range: −2 to + 2; [35]). The Pearson correlation coefficient was used to examine the bivariate association of the research variables in the current study. In multivariate analysis, a 3-step hierarchical regression was run to evaluate the effects of hope and spiritual well-being on health anxiety after controlling for demographic variables. Sociodemographic variables (age, gender, and education status) were entered in Step 1 of the analysis, which was followed by the addition of the hope score in Step 2. Spiritual well-being was added in Step 3. Data including R^2^, R^2^ changes, F changes, standardized regression coefficients (β), and *p* values were provided for each step in the regression models. In addition, to check for multicollinearity, variance inflation factor values (VIF > 4; [36]) were used. The significance level for all tests was set at 0.05.

## 3. Results 

### 3.1. Correlations

Correlations among variables are shown in Table 1. The results revealed that patients lower in total score of spiritual well-being generally had more health anxiety (*r* = −0.34, *p* < 0.01). As hypothesized, the religious spirituality (*r* = −0.40, *p* < 0.01) and existential spirituality (*r* = −0.18) subscales were negatively correlated to the outcome variable of health anxiety. 

Pearson’s correlation coefficient indicated that low scores on hope were significantly associated with more health anxiety (*r* = −0.44, *p* < 0.01). The agency subscale was negatively and significantly related to health anxiety (*r* = −0.48, *p* < 0.01). Moreover, pathway scores were significantly correlated with health anxiety (*r* = −0.33, *p* < 0.01). 

The skewness and kurtosis showed that all variables were within acceptable ranges (−2 to +2; Byrne, 2010), confirming that the variables were normally distributed (Table 1). Examining variance inflation factor (VIF) values for all predictors across all linear regression models failed to reveal any multicollinear predictors (i.e., VIF > 4; Garson, 2012), indicating that multicollinearity was not a concern for the independent variables used in this analysis (Table 2 and Table 3).

For study variables, the assumptions of multiple regression analysis were first tested. According to the Kolmogorov–Smirnov test results, the data are distributed normally. There was no multicollinearity, and the dependent and independent variables were linearly correlated.

### 3.2. Hierarchical Multiple Regression Analysis for Hope and Spiritual Well-Being Subscales as Predictors of Health Anxiety

The results of the multiple hierarchical regression analyses for hope and spiritual well-being subscales as predictors are provided in Table 2. The results indicated that at step 1, demographic variables explained 4% of the variance in health anxiety. After entry of agency and pathway at step 2, the total variance explained by the model was 33%. Therefore, agency and pathway explained 29% of the variance (R^2^ change). Adding religious spirituality and existential spirituality as predictors in step 3, the total variance in health anxiety explained was 39%. Therefore, religious spirituality and existential spirituality explained 5% of the variance in outcome (R^2^ change), and the overall model was significant (F = 3.90, *p* < 0.05). In the final model, among the predictor variables, agency (β= 0.58, *p* < 0.01) and religious spirituality (β = 0.28, *p* < 0.01) were shown to significantly negatively predict health anxiety level, but pathway and existential spirituality were not significant predictors.

### 3.3. Hierarchical Multiple Regression Analysis for Total Scores of Hope and Spiritual Well-Being as Predictors of Health Anxiety

The regression analysis for total scores of hope and spiritual well-being as predictors revealed at step 1, age, gender, and education status accounted for 4% of the variance. Adding hope as a predictor in step 2 explained 29% of the variation in health anxiety. Therefore, hope explained 25% of the variance in outcome (R^2^ change). After entry of spiritual well-being at step 3, the total variance explained by the model as a whole was 32%. Therefore, spiritual well-being explained only 3% of the variance (R^2^ change), and the overall model was significant (F = 4.44, *p* < 0.03). In the final model, both hope (β = 0.422, *p* < 0.01), and spiritual well-being (β = 0.20, *p* < 0.05) were significant predictors of health anxiety (see Table 3).

## 4. Discussion

The results showed that there is a significant negative relation between spiritual well-being and health anxiety in patients with advanced CAD. Similarly, other researchers found the same results [18,19,37,38]. In the present study, religious spirituality was more strongly associated with diminished health anxiety than was existential spirituality. A possible explanation for this could be that existential well-being during CAD may be threatened as the patient questions the meaning and purpose of life and experiences diminished life satisfaction. A review of research evidence shows that spiritual well-being has a positive impact on health and is associated with a reduction in psychological symptoms associated with different diseases [39,40]. Regarding the explanation of this finding, it can be said that spiritual well-being is a basic component of the well-being dimensions; other dimensions of well-being include physical, mental, and social well-being. Spiritual well-being is a determining and influential factor that affects the way people deal with hardships and difficulties; it creates emotions such as happiness and hope by creating meaning, a sense of purpose and self-efficacy, and a positive mental space [41]. As a result, it can reduce health anxiety in patients with CAD. 

Spiritual well-being is considered as a protective factor in promoting health and preventing diseases. Religious well-being provides social support for individuals through factors such as forging close ties with members of religious groups, establishing deep friendships with these members, and engaging in altruistic behaviors, thereby reducing stress and anxiety in individuals [42]. CAD patients with high religious well-being may cope with life challenges through reliance on the support and the guidance of God, and religious belief helps them overcome their problems. It is noteworthy that these types of coping behaviors and religious lifestyle lead to a reduction in psychological symptoms, such as anxiety in patients. Religious and spiritual beliefs play a very important role in shaping the culture of the Iranian people. In a study on a Persian sample, for example, it was shown that spirituality is a fundamental element in elderly individuals’ lives that helps them adapt to daily living conditions [43]. Therefore, the findings of the present study on the predictive role of spiritual well-being on the health anxiety of patients with CAD can also be due to these cultural aspects. In general, it can be said that spiritual well-being acts as a protective shield against health anxiety in patients with CAD and helps these patients overcome emotional distress.

Notwithstanding, the current findings indicated that hope had greater explanatory power in predicting health anxiety than did spiritual well-being as more of the variance in health anxiety was explained by hope subscales and total hope scores. The dimension of Agency was more predictive of reduced health anxiety than was the Pathway dimension of hope. Being motivated to take direct action to pursue goals was more important in giving hope than the actual pathway to achieving these goals. For patients with CAD taking the first step in cardiac rehabilitation, with a hopeful attitude that any obstacles can be overcome, may be more important than pursuing a number of alternative pathways to recovery. 

Our results on the significant negative relation between hope and health anxiety in patients with CAD is consistent with those of other researchers [18,19,20], which reported a significant relation between optimism and hope with reduced anxiety among advanced CAD patients. One study indicated that people with higher scores on the hope scale show less symptoms of depression and anxiety and have a better quality of life [44]. Moreover, anxiety was shown to be lower in people with higher levels of hope [45]. Similarly, it was suggested that health anxiety decreases with increasing hope [46]. Hope can be considered as an active and dynamic state of existential coping strategy among people with life-threatening illness [47]. Hope is a strong adaptation mechanism in patients with chronic illness, and hopeful people are better able to cope with the crisis and psychological symptoms such as anxiety. As confirmed by several studies, hope is a powerful source of coping with anxiety and stress. Hope is closely related to other positive psychological constructs, such as self-efficacy and optimism [48]. It has also been shown to be associated with promoting resilience and recovery from emotional disorders such as anxiety [48]. Therefore, it can be said that hope acts as a strong source of coping with health anxiety in patients with CAD and reduces health anxiety in these patients by potentially improving self-efficacy, optimism, and other positive psychological constructs. 

## 5. Limitations, Strengths, and Future Directions

The present study had some limitations that should be considered in generalizing and interpreting the results. Since the present research used a descriptive–correlation method, the relationships obtained from it cannot be considered as cause and effect. In addition, the study population, which included CAD patients, limits the generalization of results to other patients. However, despite the mentioned limitations, the present study enriched the research literature on the role of spiritual and psychological factors affecting the health anxiety of patients with CAD. According to the results of the present study, there is a need for health professionals to consider the therapeutic aspects of spiritual well-being and hope in order to reduce the level of health anxiety in patients with CAD.

Future research may wish to consider other means of attaining spiritual well-being such as engaging in meditation or yoga, on the health anxiety of patients with CAD. We were particularly interested in religious well-being and existential well-being as measured by the Spiritual Well-Being Scale. We acknowledge that yoga and meditation may be part of an individual’s spirituality, and for this reason, patients with recent practice of yoga or meditation were excluded. Future studies may also wish to explore gender differences in spiritual well-being and any effect this may have on health anxiety.

## 6. Conclusions

In general, it can be concluded that spiritual well-being and hope give meaning to life and lead to a positive change of attitude, increasing psychological capital and self-confidence in the life of patients with CAD, and thus are effective in reducing health anxiety in these individuals.

## Figures and Tables

**Table 1 medicina-57-01066-t001:** Means, standard deviations, range, Cronbach alphas, and intercorrelation matrix of study variables.

	Mean (SD)	Range	Skewness	Kurtosis	Alpha	1	2	3	4	5	6	7	8	9	10
1. Health Anxiety	55.96 (7.26)	37–70	−0.38	0.10	0.88	1									
2. Total spirituality well-being	62.57(5.58)	29–73	−1.93	1.48	0.86	−0.34 **	1								
3. Total hope	30.63 (5.29)	17–40	−0.32	−0.15	0.85	−0.44 **	0.40 **	1							
4. Religious spirituality	32.25 (3.22)	14–38	−1.91	1.22	0.82	−0.40 **	0.85 **	0.42 **	1						
5. Existential spirituality	30.32 (3.33)	15–37	−1.76	1.02	0.89	−0.18	0.87 **	0.26 **	0.45 **	1					
6. Agency	15.25 (3.06)	9–20	−0.55	−0.50	0.83	−0.48 **	0.45 **	0.95 **	0.43 **	0.35 **	1				
7. Pathway	15.38 (2.54)	8–20	−0.15	0.10	0.88	−0.33 **	0.28 **	0.93 **	0.35 **	0.13	0.78 **	1			
8. Age	44.53 (7.62)	23–65			-	0.06	−0.08	−0.07	−0.07	−0.07	−0.12	−0.00	1		
9. Gender	-	-			-	−0.20	0.17	0.06	0.15	0.13	0.07	0.05	−0.29 **	1	
10. Education	-	-			-	−0.05	0.01	0.19	0.02	−0.00	0.19	0.16	0.11	−0.04	1

** *p* < 0.01.

**Table 2 medicina-57-01066-t002:** Results of hierarchical multiple regressions for hope and spiritual well-being subscales as predictors of health anxiety.

Outcomes	Predictors	B	SE	Beta	T	*p*	R^2^	R^2^ Change	F Change	*P*	VIF
Health anxiety	Step 1										
	Age	0.010	0.099	0.011	0.099	0.921					1.09
	Gender	−2.949	1.576	−0.197	−1.871	0.065	0.04	0.04	1.44	0.267	1.08
	Education status	−0.445	0.630	−0.072	−0.706	0.482					1.01
	Step 2										
	Age	0.083	0.085	0.089	0.981	0.329					1.13
	Gender	−3.249	1.331	−0.217	−2.441	0.017					1.09
	Education status	−1.127	0.542	−0.181	−2.077	0.041	0.33	0.29	20.15	0.000	1.05
	Agency	−1.459	0.326	−0.622	−4.479	0.000					2.65
	Pathway	−0.250	0.386	−0.089	−0.647	0.519					2.57
	Step 3										
	Age	0.082	0.083	0.087	0.991	0.324					1.13
	Gender	−3.606	1.308	−0.241	−2.758	0.007					1.11
	Education status	−1.062	0.528	−0.171	−2.013	0.047					1.064
	Agency	−1.357	0.348	−0.578	−3.898	0.000	0.39	0.05	3.90	0.000	3.219
	Pathway	−0.405	0.390	−0.144	−1.039	0.302					2.791
	Religious spirituality	−0.616	0.221	−0.277	−2.791	0.006					1.444
	Existential spirituality	−0.213	0.211	−0.099	−1.007	0.316					1.418

**Table 3 medicina-57-01066-t003:** Results of hierarchical multiple regressions for total scores of hope and spiritual well-being as predictors of health anxiety.

Outcomes	Predictors	B	SE	Beta	T	*p*	R^2^	R^2^ Change	F Change	*p*	VIF
Health anxiety	Step 1										
	Age	0.010	0.099	0.011	0.099	0.921					1.09
	Gender	−2.949	1.576	−0.197	−1.871	0.065	0.04	0.04	1.44	0.267	1.08
	Education status	−0.445	0.630	−0.072	−0.706	0.482					1.01
	Step 2										
	Age	0.045	0.086	0.048	0.525	0.601					1.10
	Gender	−3.335	1.369	−0.223	−2.436	0.017	0.29	0.25	31.94	0.000	1.09
	Education status	−1.051	0.557	−0.169	−1.886	0.062					1.05
	Hope	−0.686	0.121	0.506	−5.651	0.000					1.05
	Step 3										
	Age	0.047	0.085	0.050	0.552	00.582					1.10
	Gender	−3.752	1.359	−0.251	−2.760	0.007	0.32	0.03	4.44	0.039	1.11
	Education status	−0.977	0.548	−0.157	−1.781	0.078					1.05
	Hope	−0.572	0.131	−0.422	−4.373	0.000					1.26
	Spiritual well-being	−0.260	0.123	−0.202	−2.106	0.038					1.24

## Data Availability

The data presented in this study are available on request from the corresponding author.

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
