# Peer review of "Associations of Spiritual Well-Being and Hope with Health Anxiety Severity in Patients with Advanced Coronary Artery Disease"

_medicina, 2021, doi:10.3390/medicina57101066_

Round 1
Reviewer 1 Report
ready to publish
Reviewer 2 Report
There are still problems with the AHS scale being referred to as the ADS scale (this was raised in the first review)
There is inconsistent use of the words 'participant' and 'subjects'. It would be good to be consistent. Generally, the word participants is preferred for humans.
Some citations are given in the text only as Name (year), e.g., line 243., while most are given as numbers in parentheses This could be edited to conform to journal format.
The article still contains text indicating that p = 0.000. It was pointed out in the initial review that that is not possible and so statistical convention is to use a < symbol, e.g. p <.001. The number of decimal places used for p values is also inconsistent throughout the manuscript.
In Tables 2 and 3 (as mentioned in my previous review of this manuscript), consistency is needed over which row the R2, R2change, F and p values are placed.
In section 3.2.and 3.3. Wouldn’t an R2 change of .04 be equivalent to 4%, not .04% as stated in the manuscript?
Line 248, the phrase 'were negatively significant' may not be a good way to describe inferential statistics. Consider rewording.
Author Response
Please check the attachment.

This manuscript is a resubmission of an earlier submission. The following is a list of the peer review reports and author responses from that submission.
Round 1
Reviewer 1 Report
well designed study and well written article. I would like to know why attending meditation and yoga are excluded. these two things in and of itself can be part of an individuals spirituality. If these subjects were in the study the existential results may have been different. In other words you may have selected a group of believers in GOD versus finding meaning in another way. this should be mentioned as a limitation. another limitation that should be mentioned is that there may be differences in male and females. We found this in our studies on our factor of reflection and introspection. should also be mentioned in limitations. (NIH HEALS and gender differences-can look up the article). Congratulations on this important work.
Reviewer 2 Report
Title
It would probably be best to avoid the abbreviation CAD in the title. Spelling it out would aid the reader’s comprehension, and also help search engines index the paper. It probably should be spelled out in the Abstract too for the same reasons.
Abstract
CAD needs to defined earlier in the article.
It may be beneficial in the abstract to include the country names, as many readers may not know where Ardabil city is.
The parenthetical abbreviations in the abstract are probably not necessary, as they are not used again in the abstract.
The low p values in the Abstract may be better written as p < .001, as technically p values of 0 are not possible. Usually the zero before the decimal point is omitted in pa values too, although the journal style may have its own rule.
The final sentence of the abstract could be weakened. It is perhaps currently a bit strong to suggest that health professionals should consider hope and spiritual wellbeing, partly because the directionality of the associations is not clear, and it’s not been shown that anxiety can be reduced.
Overall, the authors should decide whether to use ‘coronory artery disease’ or ‘CAD’ throughout the text. At the moment there is a mixture of both.
Methods,
Maybe mention that Ardibil is in Iran.
The semi-structured interview that is mentioned in the Method section, to identify eligible participants sounds like it may a bit more than screening for study recruitment. If this interview was conducted after recruitment and consent, and the questionnaires were completed, then the text is actually a description of which cases were excluded. The eligibility age range of 22 to 63 also seems a little arbitrary, unless it’s actually a description of the final sample age range (but that is given as 23-65 in Table 1). It would be good to be clear about these issues in the text. If full data was collected on more than the 100 participants described, then that might produce different results. The authors could clarify this by describing exactly how many people were screened for inclusion, and exactly how many were recruited. If there are cases that are excluded, then they should be described here, and perhaps the analyses in the results section could repeated with the full sample to confirm if the same affect is found as with the reduced N=100 sample. Arguably, a sample without excluded cases would be more representative of CAD patients.
‘Education status’ should be defined in the Methods section. It’s not until the results that the reader can see that it was an ordinal scale of highest level completed.
It is mentioned in the text that there was ethics committee approval, and that informed consent was given. Usually an ethics committee would expect participants to sign an approved consent form. It may be good to conform whether the informed consent was written (i.e., a signature on a consent form).
It is unconventional to abbreviate Adult Hope Scale to ADS, consider using AHS.
Results
The first table in the document is Table 2. Table 1 is the last one in the document. They should be placed in sequence within the text for the benefit of the reader.
In Table 2 (and Table 3), the first step, the R2 change probably should be 0.044 not 0.44, though it may be best to report only to two decimal places for consistent with similar figures in the table, i.e. 0.04.
In Table 2 it would be good to have consistency over which role the R2, and pa values are placed. For example, in Step 1 they are in the second row of that step, which is different to the other steps.
The information in section 3.1. may be better placed in the Method section.
For the data distributions of the variables, it would be better to give at least a p value to conform that all were not abnormally distributed, e.g. Kolmogorov-Smirnov tests, all p vales were greater than .05. An even better option, would be to include skew and kurtosis information fro each variable in the Table (1). Space could be saved by removing the pre decimal point zeros from the r values.
In the same paragraph, the absence of multicolinearity is reported. But this would be better examined after the regressions are reported, and again it would be better to confirm this with a brief statistic, e.g. VIF values for all models were found to be below 5, or perhaps even report all values in the tables.
In Table 3, the R2 change at the final step is only .03, and yet it is reported as being a highly significant change in the text (p = 0.000) Line 235. That seems unlikely that such a small change would be significant. Perhaps I have misunderstood, and the authors can explain, but it feels to me that the results should be checked for accuracy. Either way, it would be good to perhaps include an indication of p values for R2 change within the tables.
Discussion
Lines 241-242 there is some over size font in use. These may an issue with reference software as it appears elsewhere in the text.
It would be good to have citations for the assertion made on line 250-251.
The Discussion would be enhanced if it focused more on the current results. At the moment it feels more like a literature review. As hierarchical regression was used, the discussion could b used to explore the implications of whether adding in extra variables in the final steps increased exploratory power, i.e. that explained something not explained by the other variables.